# Factors Driving Coordinated Development of Urban Green Economy: An Empirical Evidence from the Chengdu-Chongqing Economic Circle

**DOI:** 10.3390/ijerph19106107

**Published:** 2022-05-17

**Authors:** Sentao Wu, Xin Deng, Yanbin Qi

**Affiliations:** College of Economics, Sichuan Agricultural University, Chengdu 611130, China; 2019208009@stu.sicau.edu.cn (S.W.); dengxin@sicau.edu.cn (X.D.)

**Keywords:** coordinated development, green economy, the Chengdu-Chongqing economic circle

## Abstract

Globally, all countries regard the development of economic zones around basins as the focus and main axis of national economic construction. The economic development of basin areas must consider the constraints of environmental protection, which requires local governments to adopt a coordinated development approach to the green economy. The Chengdu-Chongqing Economic Circle is located in the upper reaches of the Yangtze River Basin and will be built into a growth pole for China’s economic growth. This paper uses the panel data of 16 cities in the Chengdu-Chongqing Economic Circle from 2005 to 2019 and measures the level of coordinated development of the green economy among cities. Using fixed effect regression models, this paper identified the factors driving the coordinated development level of the urban green economy. The results show that (1) the overall trend of coordinated development of green economy in the Chengdu-Chongqing economic circle is enhanced; (2) the pulling force includes the similarity of economic agglomeration and regional openness, the resonance force includes the similarity of traffic and industrial structure, the pushing force comes from the central city; (3) in the urban agglomeration with double centers, the cooperation and competition between two “pole” cities may coexist.

## 1. Introduction

Globally, countries have been giving priority to the development of river basins, and many important civilizations and economic centers of gravity have flourished in basin areas [1,2]. In ancient times, due to the low efficiency of people’s use of environmental resource elements, several river basins, such as the Nile river basin, the Ganges river basin, and the Yangtze and Yellow river basins, which were rich in environmental resources and element resources, gave birth to the four major civilizations, the Babylonian, the ancient Egyptian, ancient Indian, and ancient Chinese civilizations [3,4,5,6]. In modern times, industrial-intensive and urban-intensive regions in many countries are also distributed along river basins [7]. For example, several industrial areas are located in the Rhine River Basin in Germany [8], and the Mississippi River Basin and the Great Lakes Basin in the United States have built a number of industrial corridors and industrial belts [9,10]. The protection of the ecological environment of the basin is closely related to the rise and fall of civilizations and the level of economic development in the basin [10]. However, the damage of human economic activities to the ecological environment of the basins is obvious [11]. How the economy of the basin can be developed while reducing the damage to the environment is a topic worthy of attention.

The Yangtze River Basin is rich in environmental and element resources [12]. The Yangtze River is the world’s largest hydropower river and the third-longest river [13], originating from the “roof of the world”, the Tanggula Mountains on the Qinghai-Tibet Plateau [14]. The Yangtze River Basin has a population of approximately 400 million people, and the output of grain, cotton, and oil from the basin accounts for more than 40% of China’s output [15]. The urban density is 2.16 times the average density of China and it is the economic zone with the highest economic density in China after the coastal areas [16]. The Yangtze River and the Yellow River are known as the mother rivers of China [17]. The Chengdu-Chongqing Economic Circle (CCEC) is located in the upper reaches of the Yangtze River Basin and is one of the most important urban agglomerations in the Yangtze River Basin [18]. The construction of the CCEC forms an important part of China’s implementation strategy of the Yangtze River Economic Belt and the Belt and Road Initiative. According to China’s designated “CCEC Ecological Environmental Protection Plan”, the local governments of CCEC will focus on the coexistence and common ecological and environmental issues of the two places, and coordinate to carry out environmental governance such as transboundary rivers and air pollution. Specifically, the concentration of PM2.5 in cities at the prefecture level and above has dropped by more than 13%, and the proportion of water quality in the state-controlled section reaching or better than Class III has reached 96%. According to the “Outline of the Construction Plan for the CCEC” issued by the State Council in 2020, China will build the CCEC as the fourth pole of China’s economic growth. The three economic growth poles currently in China are the Pearl River Delta, the Yangtze River Delta, and the Beijing-Tianjin-Hebei region [19]. The difference between the CCEC and the current three economic growth poles is that the CCEC is the only one located in the inland non-coastal region. Studying the environmental protection and economic development of the CCEC has certain reference value for the development of the world’s inland watershed areas.

Environmental protection and economic development are the two main themes of the construction of the CCEC, which requires the economic development model of the CCEC to consider the requirements of green development [20]. Green development was emphasized and considered in the process of economic growth, environmental protection, and resource conservation [21]. “Green economy” is a new economic development model that is expected to flourish under the framework of green development [22]. The green economy mainly contains two connotations: (1) Economic entities need to consider the impact of negative externalities of economic activities on the environment and its bearing capacity. This is in stark contrast to the conventional model of extensive economic development that is characterized by the destruction of the ecological environment and accompanied by high energy consumption and damage to human health [23]. (2) A green economy can obtain economic benefits and promote economic growth while protecting the environment. Currently, the Chinese government is committed to achieving the goals of carbon peaking and carbon neutralization [24,25]. The research being conducted on China’s green economic development model can serve as a reference for other developing countries to realize economic transformation and a green future.

Nurturing a green economy is an important means of realizing the sustainable growth of the global economy [26]. To realize the green economy in the urban agglomeration, the cities in the urban agglomeration are required to adopt cooperative and non-confrontational economic and environmental policies, that is, to implement the coordinated development of the green economy [27]. The coordinated development of a green economy in urban agglomerations is an important part of the construction of a double city economic circle in the CCEC. Studying the development path of the CCEC can help solve the problem of unbalanced development in underdeveloped areas.

In summary, this study contributes to the realization of a green future for the CCEC. Specifically, the focus is on the ecological environment and economic development of watersheds in underdeveloped areas. Additionally, this paper studies the coordinated development of the city level from the perspective of a green economy. Lastly, this paper examines the factors driving the coordinated development of the CCEC in the past 15 years from the perspective of the coordinated development of a green economy. The results of this study will not only elucidate the periodic changes that occur over the process of collaborative development, but will also shed light on the mechanisms of influencing factors that drive these changes.

## 2. Theoretical Analyses and Research Hypotheses

### 2.1. Green Economy and Coordinated Development

Currently, the coordinated development of a green economy is an important means of realizing the development of a green economy in real conditions. The formation of a green economy requires local governments to consider the constraints of environmental protection and resource conservation while developing the economy [28]. However, the race for economic growth prompts local governments to choose short-term economic performance with measurable and swift results. The extensive economic development model, characterized by the destruction of the ecological environment, involves high energy consumption and damage to human health [29]. The competition for development between local governments will make the boundary area form a “pollution paradise” [30] and further aggravate the environmental damage caused to the region. Therefore, to realize a regional green economy, it is paramount that the local governments adopt the idea of cooperative development and jointly realize a regional green economy.

Currently, the coordinated development of urban agglomerations has become the leading force of sustainable economic and social development globally. The coordinated development of urban agglomerations involves the dismantling of administrative divisions between two or more cities in the region, so that development factors and resources may flow freely. Such optimized resource allocation promotes the integration of economic and social development while providing complementary advantages to both cities [31]. From the perspective of synergy theory, an urban agglomeration is a complex system with self-organization abilities that can promote processes to move from disorder to order, and from low-level order to high-level [32]. This is accomplished through the interaction and integration of various subsystems within the system to realize the coordinated development of urban agglomeration. As the synergy theory emphasizes the self-organization ability of complex systems itself, it is not conducive to our understanding of the process of coordinated development of urban agglomerations. To further understand the collaborative development process of urban agglomeration, Rene Thom proposed the “catastrophe theory” [33] to study the discontinuities approaching the critical point [34,35]. Some factors driving the process of coordinated development can even produce a “chronic deposition effect” through long-term gradual change. When reaching a certain critical threshold, this type of deposition effect will cause the factors driving the coordinated development of urban agglomeration to mutate [36]. Studying the transition from gradual change to mutation is helpful to clarify the mechanism of driving factors. The formation of a green economy is based on the coordinated development of urban agglomerations and the addition of constraints, such as environmental protection and resource conservation.

### 2.2. Driving Factors of Coordinated Development of Urban Agglomeration

The “push-pull theory” of population migration alludes to the factors that attract immigrants, with the desire to improve their lives, to reside in new places [37,38]. In line with the coordinated development of urban agglomerations, the pulling force measures the pulling effect of a relatively “strong city” on a “weak city” [39]. In other words, if there is a wide gap between the two cities, the “strong city” exerts a pulling effect on the “weak city” [40].

The concept of “resonance” derived from the study of physics refers to the amplitude of a physical system which increases when the system is subject to a frequency close to the natural frequency of the system [41]. This analogy also applies to the coordinated development of urban agglomerations [42,43]. In the context of this paper, this study considers resonance to be the factor that promotes the coordinated development of the two cities due to the high similarity of some influencing factors.

“Pull” measures the upward supporting effect of a specific city on the coordinated development of any two cities [44]. The theory of “center-periphery” unbalanced development divides urban agglomerations into two parts: the center and the periphery [45,46]. The coordinated development between the peripheral cities in the urban agglomeration will be inevitably affected by the central city [47,48]. When the central city exerts a positive impact on the coordinated development of the adjacent cities [49,50], it leads to “pull.”

Collectively, the factors driving the coordinated development of a green economy can be divided into three categories: pull, resonance, and lift.

**Hypothesis** **1:**
*If the “pull” driving factor is in play, the lower the similarity of the influencing factors between the two cities, the higher the degree of collaborative development.*


**Hypothesis** **2:**
*If the “resonance” driving factor is in play, the higher the similarity of the influencing factors between the two cities, the higher the degree of collaborative development between them.*


**Hypothesis** **3:**
*If the “push” driving factors are in play, the higher the level of central city development, the higher the degree of collaborative development.*


## 3. Measurement and Space-Time Analysis of Coordinated Development

### 3.1. Data Sources

The planning outline for the construction of the CCEC was approved by the central government. It stipulates that the CCEC include 16 cities, namely Chongqing (CQ), Chengdu (CD), Zigong (ZG), Luzhou (LZ), Deyang (DY), Mianyang (MY), Suining (SN), Neijiang (NJ), Leshan (LS), Nanchong (NC), Meishan (MS), Yibin (YB), Guang’an (GA), Dazhou (DZ), Ya’an (YA), and Ziyang (ZY). This paper used the data of the 16 cities in the CCEC, from 2005 to 2019, to constitute the balanced panel data. The data in this paper are traced back to 2005, because in 2005, China included the CCEC in the National Eleventh Five-Year Plan for the first time, marking the importance of the coordinated development of the CCEC at the national level. The data in this paper are from the Sichuan statistical yearbook, Chongqing statistical yearbook, and the statistical yearbooks of various associated cities.

### 3.2. Measurement Method

#### 3.2.1. Evaluation Index System and Measurement Methods of Urban Green Economy Development (GED)

Current research suggests that the index system for evaluating the development level of a green economy has not been unified. In recent years, research on coordinated development has introduced the theory of complex systems, which holds that regional coordinated development is in itself a complex system [51]. In his book titled “Advanced System Dynamics,” Lingyi emphasized that complex systems generally have at least four subsystems that influence each other and form a closed-loop [52]. On the division of subsystems, the academic community agrees that a green economy consists of at least three subsystems: economy, society, and ecology [53]. At present, any differences in the evaluation index system of GED lie in subsystems which constitute the evaluation index system. By adding natural capital to the index system for evaluating the development level of a green economy, Jiyuan emphasized the need to consider the influence of natural resources during the evaluation of the development level of a green economy [54]. Linlin joined the two subsystems of environmental quality and governance regulation to consider the strength and effectiveness of the government’s environmental governance [55]. The green development evaluation index system issued by OECD [56] and the sustainable development index system proposed by CSD are also of importance [57]. Based on Linlin’s recommendation of an ecological coordination index system of regional complex systems, this paper establishes the complex system of the urban green economy, which includes five subsystems: economic growth, social development, environmental quality, ecological health, and policy response. The indicators in the subsystem were combined with the Chengdu-Chongqing Urban Agglomeration Development Plan issued by the National Development and Reform Commission. For more information on the index system developed to realize goals of ecological civilization and green development, see Table 1.

This paper employed the entropy weight method to measure the GED. The entropy weight method is commonly used in the academic circle to measure the development level [58,59] and is a difference-driven objective weighting method, which reflects the objective information contained in the specific evaluation data, making the determination of the index weight more objective and reasonable [59,60]. The calculation of the entropy weight method is shown in the appendix.

#### 3.2.2. Estimation Method of the Coordination Degree of GED

This paper used the physics-based capacity coupling coefficient model to measure the coupling and degree of coordination of GED between two cities [61]. The coupling model involving only two subsystems is as follows:(1)$$C_{12}^{t}=\frac{2\sqrt{U_{1}^{t}U_{2}^{t}}}{U_{1}^{t}+U_{2}^{t}}$$
where $U_{1}^{t}$ refers to the GED of city 1 in *t* year, $U_{2}^{t}$ refers to the GED of city 2 in *t* year, and its distribution interval is set to [0,1], which is calculated according to the entropy weight method of the green economy evaluation index mentioned above.

$C_{12}^{t}$ refers to the degree of coupling between cities 1 and 2 in year *t*. The value interval is set to [0,1], such that the higher the C value, the smaller the degree of dispersion and the higher the degree of coupling. In contrast, the coupling degree of the subsystems is lower. The degree of coupling can only reflect whether the two agents are similar or not, but not whether they are of low or high quality. When the $U_{1}^{t}$, $U_{2}^{t}$ values are similar and low, the degree of collaborative development presents a high value of pseudo evaluation. Therefore, to accurately reflect the coordinated development level of the GED between the two cities, it is necessary to further build a coupling coordination degree model as follows:(2)$$D_{12}^{t}=\sqrt{C_{12}^{t}\times T_{12}^{t}}$$
(3)$$T_{12}^{t}=\alpha U_{1}^{t}+\beta U_{2}^{t}$$
where $D_{12}^{t}$ refers to the coupling co-scheduling of cities 1 and 2 in *t* years, $T_{12}^{t}$. To reflect on the synergy effect on the coordinated development level of the two cities in *t* year, the undetermined coefficients α and β must meet the following requirement:(4)α+β=1

At this stage, the determination of the coefficient is controversial, and some scholars will assign both α and β a value of 0.5 each [62,63,64]. However, this method considers that the two subjects that are coupled and coordinated exert the same influence, which is inaccurate and biased. To circumvent this problem, some scholars subjectively assign values according to the importance of subjects or use correlation analysis to calculate the undetermined coefficients [65,66,67]. By combining the above methods, this study uses the ratio of urban GDP to the total GDP of the two cities to assign values to α and β.

#### 3.2.3. Analysis of the Measurement Results of the Level of Coordinated Development of a Green Economy

Based on the calculation results of a green economy, this study decided to use the coupling coordination degree model according to (1)–(4) to calculate the coupling coordination degree among cities in the CCEC from 2005 to 2019. The results of the four-year modeling are shown in Figure 1. The color depth represents the level of coordinated development of the green economy, and the darker color represents higher coordinated development of the green economy.

At the coordination level, the degree of coordination of urban coupling in CCEC and the associated regional coordination development was generally on the rise. In 2005, the average coupling coordination among cities in the CCEC was 0.408. This number increased to 0.527 in 2019. This indicates that the level of coordinated development of the green economy had significantly improved. From 2005 to 2019, compared to Chongqing, Chengdu experienced higher coupling and coordination for various cities, which likely played a strong role. This development can be attributed to the other cities that belong to the same provincial administrative unit as Chengdu. Additionally, the coordinated development within Chengdu had more advantageous policies.

At the spatial level, this study selected 4 years to plot the coupling coordination values of Chengdu and Chongqing to other cities. The distribution of coordination values exhibited obvious spatial characteristics. The level of coordinated development of neighboring cities in Chengdu and Chongqing is higher than that observed in central cities. This phenomenon is recapitulated in the U-shaped curve, with Chengdu and Chongqing occupying the poles and the central cities occupying the dipping point. This observation conforms to the characteristics of coordinated development of urban agglomerations. The central cities of Chengdu and Chongqing experienced a high degree of coordinated development, while the peripheral cities suffered from poorly coordinated development. However, the CCEC is different from the unipolar urban agglomeration of two polar cities, which produces this U-shaped structure of extremes in coordinated development.

## 4. Empirical Analysis of Driving Factors

### 4.1. Empirical Model

To determine the factors that affect the coordinated development of GED, this paper assigned the coupling coordination value of the two cities as the explained variable in the previous stage, and established a double fixed effect model as follows to analyze the possible factors influencing coordinated development:(5)$$Y_{i,j}^{t}=\beta_{0}+\overset{n}{\underset{l=1}{\sum }}\beta_{l}X_{i,j}^{t}+\theta_{i,j}+\theta_{t}+\varepsilon_{i,j,t}$$
where $Y_{i,j}^{t}$ is the degree of coordinated development of GED in the two cities. $X_{i,j}^{t}$ in t year is the similarity of influencing factors between the two cities. $\theta_{i,j}$ for individual effect, $\theta_{t}$. For the time effect, $\varepsilon_{i,j,t}$ represents a random disturbance term, t indicates time, and i,j represents a city.

The influencing factors that affect the coordinated development of GED can be divided into three categories: pulling force, resonance, and push. As Chengdu and Chongqing represent the two poles in the CCEC, the coordinated development of GED in any city must account for the influence of these two cities. Therefore, the per capita GDP of Chengdu and Chongqing in that year was introduced as a factor to measure the pair of two cities, i and j, simultaneously. Model (5) is sorted to obtain model (6) as follows:(6)$$Y_{i,j}^{t}=\beta_{0}+\overset{n}{\underset{l=1}{\sum }}\beta_{l}X_{i,j}^{t}+\beta_{cd}X_{cd}^{t}+\beta_{cq}X_{cq}^{t}+\theta_{i,j}+\theta_{t}+\varepsilon_{i,j,t}$$
where $X_{cd}^{t}$ and $X_{cq}^{t}$ represent the per capita GDP of Chengdu and Chongqing, respectively. The pull and resonance actions are represented as $X_{i,j}^{t}$.

To consider the distance factor in the study of regional coordinated development [68,69,70], a scatter plot of the logarithm of the dependent variable and distance was constructed (Figure 2). The scatter plot revealed that the closer the two cities are, the higher the level of coordinated development of a green economy between the two places. Therefore, this study will consider the effect of distance in an empirical model. Studies have shown that distance may have an impact on the degree of economic agglomeration and market subject in cities [71,72]. Therefore, this study introduces the multiplication term of distance, similarity of economic agglomeration, and similarity of market subject into model (6), and obtains model (7).
(7)$$Y_{i,j}^{t}=\beta_{0}+\overset{n}{\underset{l=1}{\sum }}\beta_{l}X_{i,j}^{t}+\overset{2}{\underset{m=1}{\sum }}\beta_{m}X_{i,j}^{t}\times D_{i,j}+\beta_{cd}X_{cd}^{t}+\beta_{cq}X_{cq}^{t}+\theta_{i,j}+\theta_{t}+\varepsilon_{i,j,t}$$

### 4.2. Variable Selection

Economic agglomeration similarity, resource endowment similarity, transportation infrastructure similarity, regional openness similarity, market subject similarity, and industrial structure similarity were selected as explanatory variables. (1) Similarity of economic agglomeration: economic agglomeration, which reflects the degree of agglomeration of economic activities that occur within a city, was measured by the labor force per unit land area [73]. (2) Similarity of resource endowment: Resource endowment refers to the total wealth of natural resources owned by an economy at a certain point in time [74]. To measure resource endowment, the ratio of employees in extractive industries to the total population at the end of the year is calculated. (3) Similarity of transportation infrastructure: transportation infrastructure is measured by traffic density, which in turn is the ratio of the total mileage of roads, railways, rivers, and waterways to the land area. (4) Similarity of regional openness: Regional openness measures the government’s business environment [20], which is generally expressed in terms of foreign direct investments. However, as the complete records of foreign direct investment in Sichuan province could not be obtained, the import and export amounts were used instead. (5) Similarity of market subjects: Market subjects can be classified based on ownership of the public and non-public economies. The market subjects are measured as the ratio of the output value of the non-public economy to the total output value [75]. (6) Industrial structure similarity: An important factor influencing the regional economy of industrial structure. This study uses the proportion of the tertiary industry in GDP to measure industrial structure similarity [76].

As the values of explanatory variables are different based on the cities, the two-city level values of the driving factor were incorporated into the coupling degree model formula. Additionally, the similarity was calculated, and the variables were represented on a logarithmic scale. The description and statistical description of the main variables in this model are shown in Table 2.

### 4.3. Empirical Results and Explanations of Driving Factors

#### 4.3.1. Test of Influencing Factors

To verify whether the selected influencing factors exert an impact on the coordinated development of GED, several steps were taken. First, the entire sample was subjected to fixed effect regression, as shown in Table 3. Distance, similarity of economic agglomeration, and similarity of market subjects were regarded as DEA and DMS.

To verify whether the double fixed effect model established in this study meets the requirements of empirical analysis, this study carried out the Hausman test to select between fixed and random effects, the Wald test for heteroskedasticity, the Pesaran test for cross-sectional dependence, and a Wooldridge test for autocorrelation in panel data. The test results are shown in Panel B of Table 3. The results show that the data of this study passed the above test, which shows that model 5, model 6, and model 7 established in this study meet the requirements of the empirical test. Panel A in Table 3 reports the empirical results. The first column of Table 3 is the double fixed effect without considering bipolar city push and distance, corresponding to model (5). The second column is the double fixed effect considering the pushing effect of bipolar cities, corresponding to model (6). The third column is the double fixed effect considering both distance and bipolar city push, corresponding to model (7). The *R*^2^ of models (5), (6), and (7) are 0.705, 0.705, and 0.713, respectively, indicating that the model fits well. The variables LEA, LRE, LT, LRO, LMS, and LIS are significant at the 1% level in all models, indicating that these six influencing factors are related to the level of coordinated development between the two cities’ relationship, similar to our expectations. Simultaneously, model (7) has the highest fitting degree, indicating that it is meaningful to consider the distance among the influencing factors. The variables LCD and LCQ are both significant at the 1% level in models (6) and (7), indicating that it is necessary to consider the role of bipolar cities in the urban agglomeration. This study also changed the calculation method of the explained variables for robustness testing. The specific method is to bring the GEDs of the two cities into the model (1) for calculation to obtain the explained variables. The test results are shown in Panel C. The directions of all variables have not changed, which proves that the conclusions obtained in this paper are robust and reliable.

#### 4.3.2. Verification of Catastrophe Theory

According to Rene Thom’s sudden change theory, the influencing factors of coordinated development may display discontinuous characteristics near the critical point. To test this hypothesis, the entire sample was divided into several intervals according to time. Next, a fixed effect regression analysis was conducted on the influencing factors in each interval and evaluated based on discontinuity near the critical point by the coefficients of each influencing factor.

Before the development and promotion of the CCEC as a national strategy, the State Council approved the Regional Plan of the Chengdu-Chongqing Economic Zone in 2011. More recently, the National Development and Reform Commission and the Ministry of Housing and Urban-Rural Development jointly issued the Development Plan of Chengdu-Chongqing Urban Agglomeration in 2016. These two documents were taken into consideration as critical developmental time points before the division of the sample time into three periods: 2005–2010, 2011–2015, and 2016–2019. The fixed effect regression was conducted on these periods and the results are shown in Table 4. Within the three intervals, economic agglomeration, resource endowment, market subject, and industrial structure are significantly similar. However, when the sign of the coefficient changed and the four influencing factors were discontinuous near the connection point, the observations were in line with the catastrophe theory.

#### 4.3.3. Distinguishing the Pulling Force and Pushing Effect of Influencing Factors

To observe the pulling effect of relatively strong cities on weak cities, it is vital to be able to distinguish these cities. The measurement results of the GED revealed Chengdu and Chongqing to be “strong cities” in the CCEC, while other cities were labeled as “weak cities.” By coupling Chengdu and Chongqing with other cities, the pulling effect of “strong cities” on “weak cities” could be observed. The results are shown in Table 5.

The similarity of the industrial structure was significant at the 1% level in models (5), (6), and (7) (Table 5). As the coefficients were all positive, it indicated that the industrial structure positively correlated with the level of coordinated development of a green economy in the two cities. The higher the similarity of industrial structure in the two cities, the higher the level of coordinated development of the green economy. The similarity of economic agglomeration and regional opening passed the tests of models (5) and (6), which showed that the similarity of economic agglomeration and the regional opening was significant at the level of 1% and 10%, respectively. As the coefficient was negative, economic agglomeration and regional opening negatively correlated with the level of coordinated development of green economies in the two cities.

These findings were further supported by the upward support of a specific city to the coordinated development of any two cities. In the CCEC, only the major cities Chengdu and Chongqing play a supporting role for other cities. Hence, the current per capita GDP of Chengdu and Chongqing were incorporated into model (5) and the results are shown in Table 6. Columns I and II measure the facilitative role of Chengdu on the coordinated development of urban agglomerations. Column II adds DEA and DMS based on column I to consider the simultaneous influence of distance. Columns III and IV measure the stimulating role of Chongqing in the coordinated development of urban agglomerations. Based on column IV, DEA and DMS were added. The *R*^2^ values of 0.705, 0.713, 0.705, and 0.713, for columns I, II, III, and IV, respectively, indicated that the model fits well.

In Table 6, columns I and II show that the LCD was significant at the 1% level and that the coefficient was positive. Collectively, this indicated that Chengdu’s per capita GDP exerted a positive impact on the coordinated development level of GED in any two cities in the region. Similarly, the LCQ was significant at the 1% level and the coefficient is positive. This suggests that the per capita GDP of Chongqing had a positive impact on the coordinated development level of GED of any two cities in the region. Furthermore, the per capita GDP of Chengdu and Chongqing were found to play a positive role in promoting the coordinated development level of GED in other cities in the urban agglomeration.

## 5. Discussion of Empirical Results

Through the fixed effect analysis of the whole sample, combined with the robustness test, this study confirmed a correlation between economic agglomeration similarity, resource endowment similarity, transportation infrastructure similarity, regional openness similarity, market subject similarity, industrial structure similarity, Chengdu per capita GDP, Chongqing per capita GDP, and the explained variables.

The degree of economic agglomeration and regional openness serve as pulling forces. According to the fixed effect results of only retaining the coupling samples of Chengdu, Chongqing and other cities (in Table 5) display interesting trends. Specifically, the similarity of economic agglomeration and regional openness were significant at the 1% and 10% levels, respectively. Additionally, the coefficients were negative. This phenomenon was explained by the influence of the “strong city” on the “weak city.” The wider the gap between the degree of economic agglomeration and the degree of regional openness, the higher the level of coordinated development of the green economy in the two cities. This finding reflects the pulling effect of “strong cities” on “weak cities” in both the degree of economic agglomeration and the degree of regional openness. Hence, the first hypothesis that this study laid out was verified. This conclusion is supported by some existing studies that show that people are more willing to gather in areas with a high degree of economic agglomeration and a good degree of regional openness, and the more obvious the difference between the two places, the stronger the willingness of people to move to advantageous areas [77,78]. This also accords with the objective reality that Chengdu and Chongqing exhibit obvious advantages in the urban agglomeration in terms of economic agglomeration and regional openness. Hence, giving priority to the development of Chengdu, Chongqing, and Chengdu’s economic agglomeration and regional openness will promote the coordinated development level of a green economy in the urban agglomeration as a whole.

Transportation infrastructure and industrial structure align with the theory of “resonance.” According to the fixed effect analysis results of the whole sample (in Table 3), the similarity of transportation infrastructure and industrial structure was significant at the 1% level. The positive coefficients further indicated that the similarity of transportation infrastructure and industrial structure between the two cities positively correlated with the level of coordinated development of the green economy between the two cities. It also shows that for most cities, with similar transportation infrastructure and industrial structure, the level of coordinated development of green economy in the two cities will be higher through resonance. Thus, the second hypothesis laid out earlier was also verified. Studies have shown that good transportation infrastructure is a necessary condition for the economic development of urban agglomerations, and simultaneously, the rational layout of the industrial structure will drive the development of urban agglomerations [79,80,81]. This is the same as the real situation in the Chengdu-Chongqing area. By the end of 2020, Sichuan’s highway mileage ranked first in China, which verifies that the balanced development of transportation infrastructure within the urban agglomeration has a positive effect on the overall balanced development of the urban agglomeration. Simultaneously, the similarity of industrial structures reflects that different cities should strengthen the cooperation mechanism of related industries and optimize the flow and allocation of factors through the industrial layout. Ensuring that transportation infrastructure and industrial structure can resonate with the coordinated development of a green economy is paramount.

The per capita GDP of Chengdu and Chongqing played a supporting role for other cities in the urban agglomeration. The addition of the per capita GDP of Chengdu and Chongqing in Table 6 had a positive impact on the coordinated development level of the green economy among cities in urban agglomeration. This shows that the current per capita GDP of Chengdu and Chongqing can play an enhancing role in the coordinated development of urban agglomerations. Thus, the final hypothesis laid out in this study was also verified. However, the existence of double-center cities in the CCEC likely causes competition between the two central cities, which may affect the coordinated development of urban agglomerations. This observation is an objective reality associated with the conception of the CCEC. The regression of the fixed effect of the whole sample also verified this observation. In the fixed effect of the whole sample, as all the other cities in the CCEC except Chongqing are under the jurisdiction of Sichuan Province, the influence of Chengdu will be significantly more pronounced. Similarly, Chengdu had a positive and Chongqing had a negative coefficient of per capita GDP (Table 4). However, the development of Chongqing has not always played a negative role in the economic circle of the Chengdu-Chongqing Twin Cities. This study explored the influence of Chongqing on the coordinated development of the CCEC in the three policy periods of 2005–2010, 2011–2015, and 2016–2019 by time interval fixed effect regression. Chongqing had a negative influence on the coordinated development of the CCEC in the two policy periods of 2005–2010 and 2011–2015. During these periods, competition dominated. With the development plan of the Chengdu-Chongqing urban agglomeration jointly issued by the National Development and Reform Commission and the Ministry of Housing and Urban-Rural Development in 2016, the coordinated development of CCEC entered a new era of development. The positive coefficient of Chongqing’s per capita GDP verified that Chongqing had switched to a cooperative mode and played an upward role in the coordinated development of a green economy in the CCEC. Most of the existing studies affirm the economic driving role of central cities in urban agglomerations and believe that central cities are the main driving force for the economic development of urban agglomerations [82].

The “rush” of the transition of cities will hinder the coordinated development of urban agglomerations. During the three policy periods of 2005–2010, 2011–2015, and 2016–2019, the coefficient of Chengdu’s per capita GDP experienced a change from positive to negative. This study attempted to regress the data of Chengdu by adding the base period and changing the measurement method. This approach showed that the economic development of Chengdu in 2016–2019 negatively correlated with the coordinated development of a green economy in CCEC. This study speculates that this may be related to Chengdu’s economic “rush” as the city entered a stage of rapid development after 2016. For instance, Chengdu hosted Jianyang to build the Tianfu International Airport in 2016. In the following year, the talent settlement policy was launched. In 2021, the resident population exceeded 20 million. This type of rush in economic development is the result of regional competition and is not conducive to the coordinated development of urban agglomerations in the long run.

## 6. Conclusions and Policy Recommendations

Globally, all countries regard the development of economic zones around basins as the focus and main axis of national economic construction. The Yangtze River is the mother river of China. The CCEC is located in the upper reaches of the Yangtze River and has been established by China as the fourth pole of economic development. Research on the coordinated development of the green economy in CCEC will provide suggestions for economic development and environmental protection in underdeveloped and non-coastal areas. In this paper, the entropy weight method was employed to measure the GED level of the CCEC from 2005 to 2019. Based on the green economy measurement values of each city each year, the coupling coordination degree model was used to measure the coordinated development level of the green economy among cities in the CCEC. Finally, the coupling coordination degree was assigned as the explained variable. The influencing factors of coordinated development were divided into pulling force, resonance, and pushing force. The main conclusions are as follows. First, the overall trend of coordinated development of a green economy in the CCEC is improving. Second, the pulling force includes the similarity of economic agglomeration and regional openness, while the resonance force includes the similarity of traffic and industrial structure. The two major cities of Chengdu and Chongqing have played promising and facilitative roles. Third, in the urban agglomeration with double centers, the cooperation and competition between two “pole” cities may coexist. This will lead to the formation of a U-shaped economic structure with two “pole” cities at the highest points, and other regions in the drooping center.

Based on the above research conclusions, this paper enlightens its readers about the process of collaborative development in future works.

First, the economic development of cities must consider environmental constraints. Economic entities need to consider the impact of negative externalities of economic activities on the environment and its bearing capacity. To achieve the green development of urban agglomerations, cities are required to adopt a development model that is cooperative rather than confrontational. Future works should pay attention to the balanced development of urban agglomerations, reduce the central collapse caused by the rush of extreme cities, and avoid unbalanced development during the contemporary round of balanced development. To accomplish this, more detailed coordinated development strategies need to be introduced at the level of urban agglomeration. Additionally, economic development factors should be allocated more reasonably, and the development momentum of each city should be balanced. Simultaneously, cities should actively integrate themselves into the process of coordinated development of urban agglomerations by actively moving closer to advantageous cities, paying attention to the division of labor and cooperation, and building industrial belts in the CCEC.

Secondly, attention should be paid to the differentiated development of the similarity of economic agglomeration and regional opening under the action of a pulling force. This approach will be essential to avoid homogeneous competition. Prosperous cities, such as Chengdu and Chongqing, should continue to attract talent, absorb a high-quality labor force to form economies of scale, and expand the degree of opening to the outside world. The integration of international partners into The Belt and Road Initiative construction will be beneficial. By building itself as a node city of foreign trade, Chengdu and Chongqing can promote the green and positive development of urban agglomerations.

Third, attention should be paid to the similarity of traffic and industrial structure under the “resonance” theory. This will ensure cooperation and balance between regions and help identify areas with weak traffic infrastructure. At all times, priority should be given to the development of traffic infrastructure and the opening of the circulation channels of production factors. In terms of industry construction, strengthening industrial cooperation among cities and paying attention to the division of labor and cooperation will be essential in building industrial integration.

## Figures and Tables

**Figure 1 ijerph-19-06107-f001:**
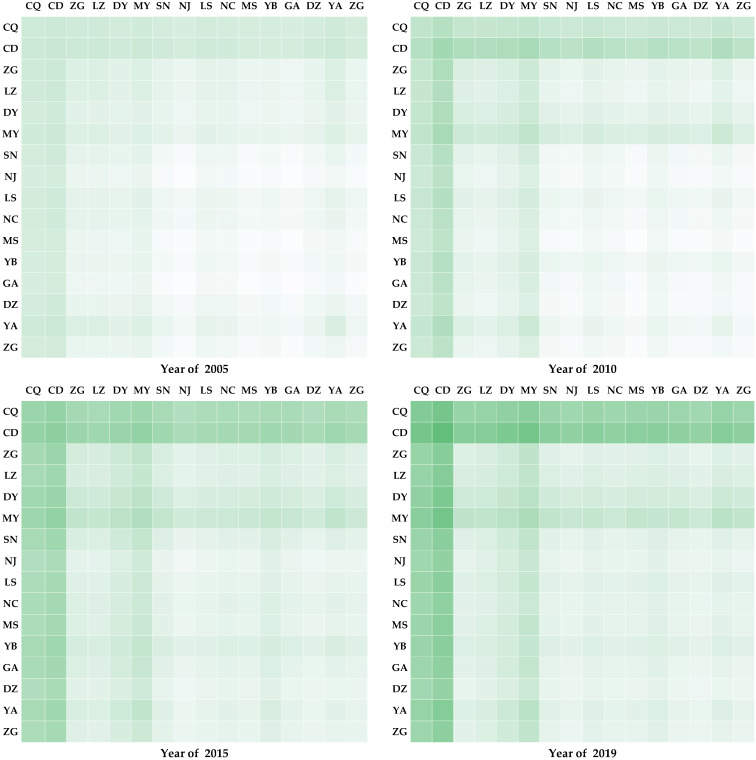
The coordinated development level of green economy of cities in CCEC in 2005, 2010, 2015, and 2019.

**Figure 2 ijerph-19-06107-f002:**
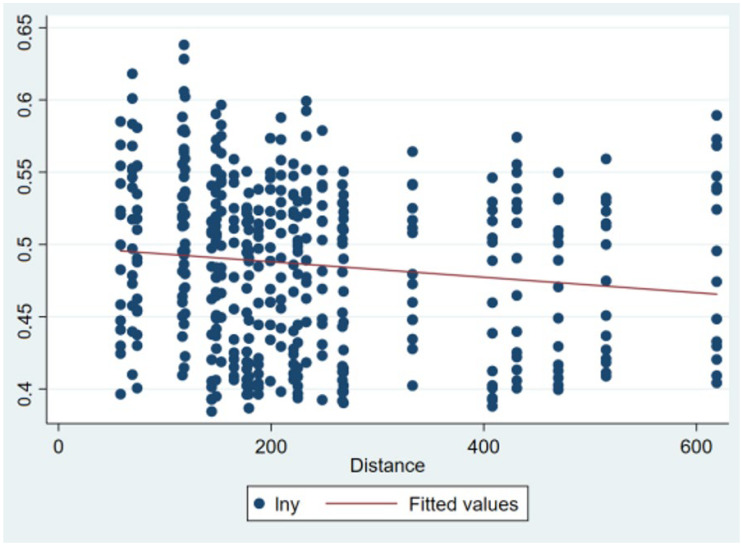
Graph dependent variable and distance scatter graph.

**Table 1 ijerph-19-06107-t001:** Evaluation index system of GED.

Order Parameter	Specific Index	Unit	Nature	Symbol
Subsystem 1: Economic growth				
Level of development	Per capita GDP	Yuan/person	Positive	X1
Structural upgrading	The proportion of tertiary industry in GDP	%	Positive	X2
Technical innovation	R&D staff equivalent to full-time staff	10,000 people/person	Positive	X3
Degree of openness	Import and export per 10,000 people	10,000 yuan/person	Positive	X4
Subsystem 2: Social development				
Urban development	Urbanization rate	%	Positive	X5
Standard of living	Engel system of urban households	%	Negative	X6
	Urban per capita disposable income	Yuan/person	Positive	X7
Input in education	Per capita financial expenditure on Education	Yuan/person	Positive	X8
Medical input	Hospital beds per capita	Bed/person	Positive	X9
Transportation	Urban road area per capita	$m^{2}$/person	Positive	X10
Subsystem 3: Environment quality				
Pollution discharge	Industrial SO2 Emissions Per capita	Tons/person	Negative	X11
	Industrial NOx Emissions per capita	Tons/person	Negative	X12
	Industrial soot and dust emissions per capita	Tons/person	Negative	X13
Subsystem 4: Natural capital				
Ecological security	Water production capacity per capita	Tons/person	Positive	X14
	Number of forest fires per capita	Times/person	Positive	X15
Green City	Green coverage rate of built-up area	%	Positive	X16
	Per capita park green space area	$m^{2}$/person	Positive	X17
Subsystem 5: Policy response				
Ecological impact	Harmless treatment rate of garbage	%	Positive	X18
	Harmless treatment rate of domestic waste	%	Positive	X19

**Table 2 ijerph-19-06107-t002:** Descriptive statistics of variables.

Variable Code	Variable Name	Average	Standard Error	Min	Max
lny	Synergy degree	0.380	0.074	0.271	0.638
Ln economic agglomeration (LEA)	Similarity of economic agglomeration	0.658	0.047	0.430	0.693
Ln resource endowment (LRE)	Similarity of resource endowment	0.546	0.144	0.154	0.693
Ln traffic (LT)	Similarity of transportation infrastructure	0.672	0.032	0.466	0.693
Ln regional opening (LRO)	Regional openness similarity	0.569	0.122	0.172	0.693
Ln market subject (LMS)	Market subject similarity	0.693	0.001	0.689	0.693
Ln industrial structure (LIS)	Industrial structure similarity	0.689	0.008	0.649	0.693
Ln Chengdu (LCD)	Per capita GDP of Chengdu	10.823	0.523	9.885	11.546
Ln Chongqing (LCQ)	Per capita GDP of Chongqing	10.475	0.595	9.420	11.236
D economic agglomeration (DEA)	Economic agglomeration and distance interaction term	3.409	0.420	2.059	4.457
D market subject (DMS)	Market and distance interaction item	3.590	0.374	2.513	4.457

**Table 3 ijerph-19-06107-t003:** Fixed effect of the whole sample.

	Model 5		Model 6		Model 7	
Panel A. Empirical Results						
LEA	−0.488 ***	(0.077)	−0.488 ***	(0.077)	−2.829 ***	(0.741)
LRE	−0.064 ***	(0.008)	−0.064 ***	(0.008)	−0.066 ***	(0.008)
LT	0.246 ***	(0.043)	0.246 ***	(0.043)	0.229 ***	(0.043)
LRO	−0.101 ***	(0.006)	−0.101 ***	(0.006)	−0.095 ***	(0.006)
LMS	−5.095 ***	(1.137)	−5.095 ***	(1.137)	−67.583 ***	(10.390)
LIS	0.434 ***	(0.121)	0.434 ***	(0.121)	0.340 ***	(0.120)
LCD			1.587 ***	(0.182)	1.568 ***	(0.179)
LCQ			−1.404 ***	(0.166)	-1.389 ***	(0.163)
DEA					0.457 ***	(0.142)
DMS					11.781 ***	(1.944)
Constant	3.816 ***	(0.804)	1.364 *	(0.824)	2.446 ***	(0.834)
Urban fixed effect	YES		YES		YES	
Year fixed effect	YES		YES		YES	
*N*	1800		1800		1800	
*R* ^2^	0.705		0.705		0.713	
Panel B. Tests						
Hausman test	225.25 ***		251.39 ***		251.39 ***	
Wald test	6894.73 ***		6894.73 ***		7394.87 ***	
Pesaran test	2.833 ***		2.833 ***		2.825 ***	
Wooldridge test	411.516 ***		660.998 ***		678.327 ***	
Panel C. Robustness checks						
LEA	−1.868 ***	(0.158)	−1.868 ***	(0.158)	−2.498	(1.531)
LRE	−0.090 ***	(0.016)	−0.090 ***	(0.016)	−0.098 ***	(0.016)
LT	0.572 ***	(0.089)	0.572 ***	(0.089)	0.539 ***	(0.088)
LRO	−0.216 ***	(0.012)	−0.216 ***	(0.012)	−0.209 ***	(0.012)
LMS	−11.045 ***	(2.341)	−11.045 ***	(2.341)	−135.845 ***	(21.482)
LIS	1.446 ***	(0.249)	1.446 ***	(0.249)	1.265 ***	(0.249)
LCD			0.960 **	(0.374)	0.945 **	(0.371)
LCQ			−0.830 **	(0.341)	−0.817 **	(0.338)
DEA					0.133	(0.294)
DMS					23.495 ***	(4.020)
Constant	7.743 ***	(1.657)	6.072***	(1.697)	8.317 ***	(1.725)
Urban fixed effect	YES		YES		YES	
Year fixed effect	YES		YES		YES	
*N*	1800		1800		1800	
*R* ^2^	0.492		0.492		0.502	

Note: Standard errors are in parentheses; ***, **, and * are significant at 1%, 5%, and 10%, respectively.

**Table 4 ijerph-19-06107-t004:** Fixed effect of time-interval.

	2005–2010			2011–2015			2016–2019		
	Model 5	Model 6	Model 7	Model 5	Model 6	Model 7	Model 5	Model 6	Model 7
LEA	−0.746 ***	−0.746 ***	−1.950 ***	0.414 **	0.414 **	−1.250	−0.792 *	−0.792 *	−8.050 *
	(0.064)	(0.064)	(0.609)	(0.167)	(0.167)	(1.807)	(0.449)	(0.449)	(4.180)
LRE	−0.018 **	−0.018 **	−0.018 **	0.008	0.008	0.008	0.030 ***	0.030 ***	0.032 ***
	(0.007)	(0.007)	(0.007)	(0.007)	(0.007)	(0.007)	(0.010)	(0.010)	(0.010)
LT	0.020	0.020	0.023	−0.742 ***	−0.742 ***	−0.738 ***	−0.394 ***	−0.394 ***	−0.398 ***
	(0.026)	(0.026)	(0.026)	(0.094)	(0.094)	(0.094)	(0.112)	(0.112)	(0.112)
LRO	−0.019 ***	−0.019 ***	−0.018 ***	−0.031 ***	−0.031 ***	−0.031 ***	−0.006	−0.006	−0.006
	(0.006)	(0.006)	(0.006)	(0.008)	(0.008)	(0.008)	(0.009)	(0.009)	(0.009)
LMS	0.682	0.682	5.104	10.754 ***	10.754 ***	−51.987 *	−6.337 ***	−6.337 ***	−14.124
	(1.056)	(1.056)	(9.224)	(3.580)	(3.580)	(31.180)	(1.291)	(1.291)	(11.742)
LIS	−1.625 ***	−1.625 ***	−1.608 ***	0.621 **	0.621 **	0.554 **	1.529 ***	1.529 ***	1.483 ***
	(0.140)	(0.140)	(0.140)	(0.243)	(0.243)	(0.244)	(0.202)	(0.202)	(0.205)
LCD		0.514 ***	0.513 ***		0.035 ***	0.035 ***		−0.084 ***	−0.084 ***
		(0.115)	(0.115)		(0.007)	(0.007)		(0.014)	(0.014)
LCQ		−0.447 ***	−0.446 ***		0.061 ***	0.060 ***		0.298 ***	0.295 ***
		(0.104)	(0.104)		(0.005)	(0.005)		(0.029)	(0.029)
DEA			0.233 **			0.322			1.435 *
			(0.117)			(0.343)			(0.818)
DMS			−0.829			11.476 **			1.508
			(1.726)			(5.661)			(2.204)
cons	1.488 **	0.618	0.521	−7.272 ***	−8.284 ***	−5.974 **	4.509 ***	2.159 **	2.072 **
	(0.725)	(0.734)	(0.748)	(2.433)	(2.433)	(2.685)	(0.957)	(0.961)	(0.978)
Urban fixed effect	YES	YES	YES	YES	YES	YES	YES	YES	YES
Year fixed effect	YES	YES	YES	YES	YES	YES	YES	YES	YES
*N*	720	720	720	600	600	600	480	480	480
*R* ^2^	0.633	0.633	0.636	0.911	0.911	0.912	0.643	0.643	0.647

Note: Standard errors are in parentheses; ***, **, and * are significant at 1%, 5%, and 10%, respectively.

**Table 5 ijerph-19-06107-t005:** Distinguishing the fixed effect of pulling force.

	Model 5	Model 6	Model 7
LEA	−0.143 ***	−0.143 ***	−0.721
	(0.047)	(0.047)	(0.465)
LRE	−0.001	−0.001	−0.003
	(0.006)	(0.006)	(0.006)
LT	−0.204 ***	−0.204 ***	−0.230 ***
	(0.049)	(0.049)	(0.049)
LRO	−0.011 *	−0.011 *	−0.010
	(0.006)	(0.006)	(0.006)
LMD	3.013 ***	3.013 ***	−25.041 ***
	(0.790)	(0.790)	(7.486)
LIS	0.619 ***	0.619 ***	0.556 ***
	(0.117)	(0.117)	(0.119)
LCD		1.056 ***	1.117 ***
		(0.186)	(0.184)
LCQ		−0.871 ***	−0.927 ***
		(0.170)	(0.168)
DEA			0.115
			(0.090)
DMS			5.175 ***
			(1.372)
Constant	−1.879 ***	−4.114 ***	−3.672 ***
	(0.556)	(0.590)	(0.593)
Urban fixed effect	YES	YES	YES
Year fixed effect	YES	YES	YES
*N*	420	420	420
*R* ^2^	0.978	0.978	0.979

Note: Standard errors are in parentheses; *** and * are significant at 1% and 10%, respectively.

**Table 6 ijerph-19-06107-t006:** Distinguishing the fixed effect of the pushing effect.

	I	II	Ⅲ	Ⅳ
LEA	−0.488 ***	−2.829 ***	−0.488 ***	−2.829 ***
	(0.077)	(0.741)	(0.077)	(0.741)
LRE	−0.064 ***	−0.066 ***	−0.064 ***	−0.066 ***
	(0.008)	(0.008)	(0.008)	(0.008)
LT	0.246 ***	0.229 ***	0.246 ***	0.229 ***
	(0.043)	(0.043)	(0.043)	(0.043)
LRO	−0.101 ***	−0.095 ***	−0.101 ***	−0.095 ***
	(0.006)	(0.006)	(0.006)	(0.006)
LMD	−5.095 ***	−67.583 ***	−5.095 ***	−67.583 ***
	(1.137)	(10.390)	(1.137)	(10.390)
LIS	0.434 ***	0.340 ***	0.434 ***	0.340 ***
	(0.121)	(0.120)	(0.121)	(0.120)
LCD	0.052 ***	0.051 ***		
	(0.001)	(0.001)		
DEA		0.457 ***		0.457 ***
		(0.142)		(0.142)
DMS		11.781 ***		11.781 ***
		(1.944)		(1.944)
LCQ			0.047 ***	0.046 ***
			(0.001)	(0.001)
Constant	3.307 ***	4.367 ***	3.372 ***	4.431 ***
	(0.803)	(0.814)	(0.803)	(0.814)
Urban fixed effect	YES	YES	YES	YES
Year fixed effect	YES	YES	YES	YES
*N*	1800	1800	1800	1800
*R* ^2^	0.705	0.713	0.705	0.713

Note: Standard errors are in parentheses; *** is significant at 1%.

## Data Availability

The datasets used and/or analyzed during the current study are available by reasonable request.

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
