# Peer review of "Factors Driving Coordinated Development of Urban Green Economy: An Empirical Evidence from the Chengdu-Chongqing Economic Circle"

_ijerph, 2022, doi:10.3390/ijerph19106107_

Round 1
Reviewer 1 Report
1. Remove parenthesis with references from abstract
2. Specify the reason for the choice of cities and the time frame - 2015-2019.
3. What is the reason for not having updated data until 2022?
4. Replace the first-person writing with the third person (impersonal form).
5. Improve Fig. 2 - suggest enlarging the graph.
6. Does table 2 contain a statistical analysis? What is it? Wouldn't it simply be distance data? If it is statistical data, mention it following the respective patterns of statistical variables (just like in Table 3).
7. Reading Table 4 with references to the columns in parentheses was confusing. Note the following ones as well.
8. Although the authors mentioned policy recommendations in their conclusions, I could not identify any environmental policy recommendations.
Reviewer 2 Report
The authors aim to “shed light on the coordinated development of a green economy in urban agglomerations” however I have two major concerns that I think they should be addressed before the paper can be considered for publication in International Journal of Environmental Research and Public Health.
- The first concern relies on conceptual framework of the research. Why the CCEC region above all others? Is this region the pioneer in green economy development in China? I think this importance of CCEC region in China development in green economy must be clarified in introduction section. Furthermore, the authors conclude about development of urban agglomeration. Do they think that a study on CCEC region is enough to conclude on all urban agglomerations? Why do they not apply the study to more than one region? This should be clarified in conclusion.
- The second concern relies on the methodological approach. The authors based the paper on fixed-effect regression models; however I think that there is a lack of tests that good practices dictate before select a fixed-effects regression. For instance, an Hausman test (or Robust Hausman) to select between fixed or random effects; a group-wise Wald test for heteroskedasticity; a Pesaran test for cross-sectional dependence; and a Wooldridge test for autocorrelation in panel data would be the minimum required. Without those tests it can’t be assured that coefficients are no biased.
- In the line 351 the authors refer that “We also tested the robustness of the above results.” How have they teste the robustness of the results? In fact, I see a wider model (model 3 – tables 4, 5 and 6) where authors introduce two new variables (“DEA” and “DMS”) and in consequence the magnitude of LEA and LMD coefficients increase immensely compared to model 1 and 2.
I think those points should be developed and clarified before properly conclude about their results.
The paper should also be revised for some minor fixes. For instance: (i) Why authors choose entropy weight method to measure the development level of a green economy?; (ii) in lines 216-218 the authors refers that “some scholars will assign…”. Some references are required.; (iii) The minimum and maximum of the variable should be shown in table 3 to identify and avoid outliers; (iv) confront the results with literature and raised hypothesis in the discussion section.
Round 2
Reviewer 1 Report
The authors have responded to virtually all suggestions kindly.
The time frame was justified and the graphics were implemented.
I recommend the article for publication.
Congratulations to the authors.